# Purification and Identification of Flavonoid Molecules from *Rosa setate x Rosa rugosa* Waste Extracts and Evaluation of Antioxidant, Antiproliferative and Antimicrobial Activities

**DOI:** 10.3390/molecules27144379

**Published:** 2022-07-08

**Authors:** Mengqi Wu, Jingying Xu, Hui Zhang, Wei Xia, Wei Li, Wenqing Zhang

**Affiliations:** 1Shanghai Key Laboratory of Functional Materials Chemistry, School of Chemistry and Molecular Engineering, East China University of Science and Technology, Shanghai 200237, China; wumengqi@ecust.edu.cn (M.W.); xujingying0430@126.com (J.X.); nicaia10@163.com (H.Z.); xiawei1999@ecust.edu.cn (W.X.); 2State Key Laboratory of Dairy Biotechnology, Shanghai Engineering Research Center of Dairy Biotechnology, Dairy Research Institute, Bright Dairy & Food Co., Ltd., Shanghai 200436, China

**Keywords:** *Rosa setate x Rosa rugosa*, waste biomass, purification, flavonoids, antioxidant, antiproliferative, antimicrobial

## Abstract

*Rosa setate x Rosa rugosa* is widely used in the essential oil industry and generates large amounts of waste annually. The purpose of this research is the recycling of bioactive flavonoids from rose waste biomass to develop high-value products. Resin screening and adsorption/desorption dynamic analysis showed that HP20 resin was suitable to purify the flavonoids from *R. setate x R. rugosa* waste extracts. Under the optimal enrichment process, the product had a 10.7-fold higher purity of flavonoids with a satisfactory recovery of 82.02%. In total, 14 flavonoids were identified in the sample after purification by UHPLC-QTOF-MS. Moreover, the DPPH and ABTS assays revealed that the flavonoids-purified extracts exhibited higher antioxidant activities than the crude extracts. Meanwhile, the purified extracts presented stronger antiproliferative activity against HepG2, Caco-2, MCF-7 and A549 cell lines. The bacteriostatic effects of the purified extracts against four bacteria (*Staphylococcus aureus (S. aureus)*, *Escherichia coli (E. coli)*, *Staphylococcus epidermidis (S. epidermidis)*, *Pseudomonas aeruginosa (P. aeruginosa*)) and yeast (*Candida albicans (C. albicans*)) were stronger compared with the crude extracts. It was concluded that flavonoids-enriched extracts from *R. setate x R. rugosa* waste had the potential to be applied in functional food and pharmaceutical industries.

## 1. Introduction

*Rosa Setate x Rosa Rugosa* (Kushui Rose) is an important industrial crop native to China, belonging to the Rosaceae family [1]. Due to its pharmacological properties, such as antioxidative, anti-inflammatory, anticancer effects, rose used to be a traditional Chinese medicine for infections, chronic pain, diarrhea, inflammatory diseases, and women’s health for thousands of years [2,3,4]. Nowadays, the main processing product of rose petals is volatile oil, which is widely used in the cosmetic industry. The production of rose essential oil on an industrial level leads to a considerable quantity of solid wastes, which poses serious environmental issues. Nevertheless, it has been demonstrated that they still contain exploitable natural sources of bioactive compounds, such as flavonoids, polyphenols and polysaccharides. Previous studies have suggested that rose waste materials are promising sources of biomacromolecules. For example, a neutral polysaccharide WSRP-1b isolated from *R. Setate x R. Rugosa* waste had impressive immunomodulatory activities [5], and another two pectic polysaccharides with different structure characteristics both exhibited hypoglycemic and immunomodulatory activities [6]. It was also reported that an immunostimulatory pectic polysaccharide was isolated from essential oil industrial wastes of *Rosa damascena* [7]. In addition, Slavov et al. reported a technical scheme for the combined recovery of polysaccharides and polyphenols from *Rosa damascena* Mill. wastes [8].

Flavonoids, characterized with the basic skeleton of three rings (C6-C3-C6), are the main class of polyphenolic compounds produced in plants as secondary metabolites. They have shown health benefits with anticarcinogenic, antioxidative, antihyperglycemic, and neuroprotective properties [9,10,11,12]. As the purity of flavonoids from plants is reportedly positively related to their biological activities [13], it is crucial to establish an efficient method to improve flavonoids purity. At present, many techniques for the enrichment of flavonoids are inefficient and not suitable for large-scale industrial production as they are time- and solvent-consuming, such as ultrafiltration, supercritical fluid extraction, high-speed counter-current chromatography [14], aqueous two-phase extraction [15], molecularly imprinted membrane [16] and preparative HPLC [17,18,19]. Because of their relative ideal pore structure, various polarity ranges, high efficiency, environmental friendliness, energy conservation and easy operation, macroporous resins are widely used in flavonoids enrichment from many crude extracts of natural resources such as *Pteris ensiformis* Burm. [20], *Brassica oleracea* L. var. *acephala* L. [21] and *Olea europaea* L. (olive) leaves [22]. Therefore, macroporous adsorption resins are suitable for enrichment of flavonoids from *R. Setate x R. Rugosa* waste extracts.

To the best of our knowledge, there have been no previous reports on purification and biological activities of flavonoids from Kushui rose waste extracts. Therefore, the goals of this work were to: achieve effective recovery of high-purity flavonoids of Kushui rose waste biomass using MAR column chromatography; analyze and identify flavonoids in the purified extracts using UHPLC-QTOF-MS technology; and comparatively investigate the antioxidant, antiproliferative and antimicrobial activities of the crude extracts and purified extracts in vitro. The results may provide a fundamental theoretical basis for integrated utilization and contribute to further application in functional food and pharmaceutical areas of *R. Setate x R. Rugosa* waste.

## 2. Results and Discussion

### 2.1. Static and Dynamic Screening of MARs

Eleven resins were screened for the adsorption/desorption characteristics of total flavonoids (TFs) in *Rosa setate x Rosa rugosa* waste extracts, which are largely related to resin properties, such as surface polarity, specific area and pore diameter. The results of Figure 1 indicate that the adsorption capabilities of eleven MARs towards flavonoids were very close, in the range of 69.61–83.24 mg/g, while the desorption capacities of HP20, HPD-500 and HPD-300 resins were 72.42, 71.63 and 68.08 mg/g, respectively, i.e., higher than the other MARs. Although the adsorption and desorption capacity of HP20 and HPD-500 were close, it was found that the desorption ratio of HP20 (90.88%) resin was higher than HPD-500 (88.47%). This may be due to the steric effect such as the size and pore diameter of MARs, which could facilitate the desorption process [23]. HP20 and HPD-500 resins were selected by comprehensive analysis for the following dynamic tests. As shown in Appendix A, the recovery rate (88.13%) and purity (42.42%) of HP20 resin towards flavonoids were higher than HPD-500. Consequently, HP20 resin was considered as the most suitable adsorbent for further enrichment of TFs in Kushui rose waste. It is widely reported that HP20 resin can be chosen to recover natural products from plant materials [24,25].

### 2.2. Resin Column Chromatography

#### 2.2.1. Dynamic Breakthrough Curves

The amount of adsorbent, the feed rate and the feed volume in a fixed-bed column system have an impact on the saturated adsorption capacity. Hence, it is essential to set up dynamic breakthrough curves to obtain the optimal feed conditions of a sample. Generally, when the concentration of TFs in eluent reaches 10% of the initial sample concentration, it is called the breakthrough point [26]. As displayed in Figure 2A, the lower the flow rate, the better the adsorption property towards flavonoids. This could be explained by the prolonging of flavonoids–resin contact time, which could positively affect the diffusion in particles, which was beneficial to improve the adsorption capacity. When reaching the breakthrough point, the feed volumes were 4, 3, 2, and 1.5 BV, respectively, at feed rates of 2, 3, 4, and 5 BV/h. For economical consideration, 4 BV (72 mL) was selected as the feed volume corresponding to 2.0 BV/h of flow rate.

#### 2.2.2. Effect of Ethanol Concentration

An excellent eluent should not only elute the adsorbate easily and effectively, but also needs to be non-toxic and environmentally friendly. Hence, the TFs concentrations in each fraction elute 10 BV d by various concentrations of ethanol at the flow rate of 2 BV/h were monitored. As Appendix A shows, the purity of flavonoids increased from to 14.92% to 48.11% with increase of ethanol concentration, reaching a maximum value at 30% ethanol and then showing a decline trend. Some studies have confirmed that flavonoids have poor solubility in lower ethanol concentrations, while some alcohol-soluble impurities could be competitively dissolved in higher concentrations of ethanol [27,28]. Thus, 30% (*v/v*) ethanol was regarded as the suitable eluent.

#### 2.2.3. Dynamic Desorption Curves

Dynamic desorption curves of HP20 were obtained by eluting with 30% ethanol at different flow rates of 2, 3, 4 and 5 BV/h, which can represent how elution speed rate and volume affect the desorption ratio. As demonstrated in Figure 2B, the low flow rate had a favorable impact on desorption performance. At the elution speed rates of 2, 3, 4 and 5 BV/h, the flavonoids were almost completely eluted in the corresponding eluent volumes of 7.5, 9, 10 and 11 BV, while the recovery yield demonstrated a declining trend of, respectively, 82.86%, 82.02%, 77.39% and 73.98%. Taking into consideration operational efficiency, solvent usage and recovery yield, 3 BV/h was referred to as the appropriate elution speed rate. After this small-scale enrichment, the purity of TFs reached 50.10% in the dried product, which was 10.7-fold higher than crude extracts.

### 2.3. Identification of Flavonoids by UPLC-QTOF-MS

Liquid chromatography combined with mass spectroscopy is an extremely helpful technique for peak assignment and further identification and elucidation of the individual constituents. The purified product was analyzed by UHPLC-ESI-QTOF-MS in order to obtain a tentative identification of its flavonoids. The total ion chromatogram (TIC) of the purified extracts of *R. setate x R. rugosa* waste is given in Figure 3. The peaks in purified extracts were well separated and 14 flavonoids were identified. Data on the retention times (tR, min), accurate masses ([M−H]^−^, *m/z*), molecular formula and major MS/MS fragments of the proposed compounds are shown in Table 1. In this work, a total of 14 flavonoid glycosides, primarily quercetin and kaempferol derivatives, were identified through their retention times and mass spectral data compared with standards or the available literature [29,30,31].

In general, flavonoid O-glycosides typically exhibit the neutral losses of sugar residues as the initial cleavage. The compounds corresponding to peaks 7 and 14 (Appendix A) were identified as rutin (further confirmed via standard) and quercitrin, which had *m/z* values of 609.15 and 447.10. The aglycone ion of them both at *m/z* 301.04 [A–H]^−^ indicated the neutral losses of rutinose (−308 Da) and rhamnose residue (−146 Da), respectively. Comparing the retention times with standard compounds, the compounds corresponding to peaks 6 and 8 (Appendix A) with the same fragment ions at *m/z* 463.09 [M−H]^−^, 301.04 [A–H]^−^, 300.03 [(A-2 H)]^−^ were respectively identified as hyperoside (quercetin 3-*O*-galactoside) and isoquercitrin (quercetin 3-*O*-glucoside). Additionally, the compounds corresponding to peaks 9 and 12 (Appendix A) had the same protonated ion at *m/z* 433.08 [M−H]^−^, and the significantly abundant aglycone ion at *m/z* 301.04 [A–H]^−^ revealed a neutral loss of pentose residue (−132 Da), so were both designated as quercetin-3-*O*-pentoside. Similarly, compounds corresponding to peaks 3, 5, 11 and 13 (Appendix A) were identified as kaempferol-3,4′-di-*O*-diglucoside, kaempferol-3-*O*-sophoroside, kaempferol-3,7-di-*O*-rhamnoside and kaempferol-3-*O*-glucoside (Astragalin), respectively, with the aglycone ion at *m/z* 284 [M-2 H]^−^ in the negative ion mode, which had been previously identified by [32,33]. Meanwhile, the compounds corresponding to peak 4 (Appendix A) with a protonated ion [M-H]^−^ at *m/z* 639.16 were tentatively identified as isorhamnetin-3-*O*-sophoroside, previously reported by [34]. The fragment ions at *m/z* 477.10 and 315.05 were generated by the successive losses of glucose residue twice. The daughter ion at *m/z* 299.02 could be assigned to the continuing loss of methyl from the aglycone ion at *m/z* 314.05 [A-H]^−^.

As is well known, flavonoids and polyphenolic compounds exhibit high antioxidant activity ascribed to their function against free radicals and they are present in many plants, including medicinal herbs, fruits and vegetables [35,36]. To investigate antioxidant capacity in food or biological systems effectively, various methods specific to their chemical properties have been used. In this study, all experiments were performed to evaluate the antioxidant activity of the crude extracts and purified products, in comparison to Trolox.

As illustrated in Figure 4A, the scavenging effect for DPPH radical of the crude extract was in a dose-dependent manner (6.25–50 μg/mL), whereas the DPPH radical scavenging activity of the purified extract and the positive control (Trolox) both increased rapidly, and their highest scavenging rate reached 82.70% and 83.64% at the dose of 25 μg/mL, separately. The IC_50_ values for the scavenging effects were 7.64 ± 0.23 μg/mL and 44.6 ± 0.51 μg/mL for the purified extracts and the crude extracts, respectively, which were higher than that of Trolox (4.86 ± 0.05 μg/mL). The flavonoids-purified extracts showed significantly higher scavenging activities for DPPH radicals compared with crude extracts. It could be inferred that a positive relationship between the content of total flavonoids existed in *Rosa setate x Rosa rugosa* waste extracts and DPPH radical scavenging activity. Furthermore, a previous study confirmed that the ethyl acetate fraction of rose petal methanolic extract had an extremely high antioxidant activity potential, which depends largely on its components (most likely flavonoids) [37]. The results of this study were consistent with their findings, indicating that the *R. setate x R. rugosa* waste biomass also could be a new source of natural antioxidants.

With regard to the ABTS•^+^ assay (Figure 4B), the ABTS•^+^ scavenging activity of the crude and purified extracts significantly increased with increasing concentrations from 6.25 to 100 μg/mL. The maximum ABTS•^+^ scavenging rates of the crude and purified extracts were 93.46% and 93.07%, which were close to Trolox (93.66%), and the corresponding IC_50_ values were estimated to be 32.21 ± 0.54 μg/mL and 15.90 ± 0.78 μg/mL, respectively (IC_50_ of Trolox = 30.23 ± 0.19 μg/mL). Interestingly, among the tested samples, the flavonoids extracts from Kushui rose waste possessed the strongest ABTS scavenging activity. This might be due to the synergistic effect of flavonoids in the purified product in terms of antioxidant activity [38,39]. As these results indicated, it was very useful to develop the purification process in this research. Similarly, the flavonoids extract of Lotus (*Nelumbo nucifera*) leaves purified by D101 resin column had noticeable scavenging activity on ABTS radicals [40].

### 2.4. Antiproliferative Activity

Rose petals are known to have a potential antiproliferative activity, which can be attributed to the compounds such as flavonoids, gallic acid, protocatechuic acid and tannins in them [2,41]. Therefore, antiproliferative activities of the crude and purified extracts from *Rosa setate x Rosa rugosa* waste against liver (HepG2), colon (Caco-2), breast (MCF-7) and lung (A549) cancer cell lines were evaluated by MTT assay. Herein, the samples before and after enrichment inhibited the proliferation of four cell lines in a concentration-dependent manner within the range of 25–500 μg/mL (Figure 5). After HepG-2, MCF-7, Caco-2 and A549 cell lines were treated with samples at a concentration of 500 μg/mL for 48 h, the minimum cell viabilities for purified extracts were 10.28%, 45.02%, 39.10%, and 53.67%, respectively, while those of other cells for crude extracts were higher than 60%, except for HepG2 cells which reached 47.76%. The IC_50_ values of crude and purified extracts on four cell lines ranged from 81.43 to 422.43 μg/mL according to the variety of cells in the following increasing order: HepG2 (215.80 and 81.43 μg/mL), Caco-2 (230.54 and 119.93 μg/mL), MCF-7 (405.70 and 224.28 μg/mL) and A549 (422.43 and 253.15 μg/mL). Evidently, the flavonoids-purified extracts exhibited relatively higher antiproliferative activities against cancer lines compared with crude extracts. These results revealed that a stronger anti-hepatoma activity was found in the purified extracts, which was well correlated with their higher total flavonoid contents. This might also be related to the presence of flavonoid glycosides in the purified extracts, such as astragalin and isoquercitrin, attributed to the glucose group at C-3 in their structures [42].

### 2.5. Antimicrobial Activity

The antimicrobial activities of the sample before and after enrichment were studied in several strains of Gram-negative and positive bacteria, as well as yeast. As presented in Figure 6, the crude extract exhibited a good antibacterial activity against *S. aureus, S. epidermidis and P. aeruginosa* except for a resistant Gram-negative bacterium, *E. coli*, as well as *C. albicans*, while the purified extracts had great potential for bacteriostatic effects against all tested strains. The ZOI and MIC values of the crude and purified extracts against all the tested strains are shown in Table 2. The most sensitive microorganisms to the purified extracts were *S. aureus*, *S. epidermidis* and *P. aeruginosa* with inhibition zones of 20.7 ± 0.40, 18.90 ± 0.10 and 16.42 ± 0.22 mm, respectively. Other microorganisms were found to be less sensitive with inhibition zones ranging from 10 to 14 mm (Table 2). Additionally, the strongest antibacterial activity of the purified product was observed against *S. aureus* and *S. epidermidis* (based on their MIC values: 1.563 mg/mL). Similarly, the flavonoids extract from *Panax notoginseng* flowers purified by AB-8 macroporous adsorption resin presented obvious inhibitory effects on *S. aureus* and *P. aeruginosa* [43]. Natural flavonoids and phenolic compounds act as antibacterial agents against many pathogenic bacteria. These results indicated that the purified extract of *Rosa setate x Rosa rugosa* waste is rich in active antimicrobial ingredients, making it a potential candidate as a natural antimicrobial agent in medicinal industries.

## 3. Materials and Methods

### 3.1. Reagents and Chemicals

Rutin trihydrate, hyperoside, isoquercitrin, 2,2-diphenyl-1-picrylhydrazyl (DPPH), 2,2′-azino-bis(3-ethylbenzothiazoline-6-sulfonic acid) diammonium salt (ABTS), 2,4,6-tris(2-pyridyl)-s-triazine (TPTZ), Trolox, and 3-(4,5-dimethylthiazol-2-yl)-2,5-diphenyltetrazolium bromide (MTT) were purchased from Sigma-Aldrich Chemical Co. (St. Louis, MO, USA). UHPLC grade acetonitrile was received from Thermo Fisher Scientific (Shanghai, China). HepG2, MCF-7, Caco-2 and A549 cells were obtained from the Type Culture Collection of the Chinese Academy of Sciences (Shanghai, China). Fetal bovine serum (FBS), penicillin/streptomycin and DMEM were purchased from Gibco (Carlsbad, CA, USA). Other reagents were analytical grade and purchased from Shanghai Titan Scientific Co., Ltd. (Shanghai, China).

### 3.2. Preparation of Crude Total Flavonoid Extracts

The *Rosa setate x Rosa rugosa* wastes were collected from Kushui, Yongdeng county, Gansu Province, China (2017 harvest). The rose residues were vacuum dried at 50 °C to achieve a constant weight and sieved through a 40-mesh sieve. Then, 5.0 kg of powdered samples was extracted twice using 20-fold 75 % (*v/v*) ethanol under reflux for 2 h. After filtration, the crude extracts were concentrated to dryness for standby, storing in the dark at −20 °C.

### 3.3. Determination of Total Flavonoid Contents

The content of total flavonoids was determined by the colorimetric method with slight modifications [44]. Briefly, 1.0 mL of 5% NaNO_2_ solution was reacted with sample solution (5.0 mL in 75% ethanol). After 6 min, 1.0 mL of 10% Al(NO_3_)_3_ solution was added to the above solution. After reaction for another 6 min, 10.00 mL of 4% NaOH solution was added. The absorbance was measured at 510 nm 15 min later on a UV2600 Spectrophotometer (Shimadu, Japan). Based on the rutin standard curve in the 0.02–0.10 mg/mL concentration range, the regression equation was expressed as A = 12.4300 ρ + 0.0350, which exhibited a good linear relationship (R^2^ = 0.9999). The contents of TFs were expressed as mg rutin equivalent/g extracts.

### 3.4. Purification of Flavonoids by MARs

#### 3.4.1. Pretreatment of MARs

The physical properties and manufacturers of eleven resins are shown in Appendix A. Before experimental use, aliquots of adsorbents were leached with ethanol overnight and washed thoroughly with the distilled water as the monomers and porogenic agents in newly bought MARs would affect absorption [45].

#### 3.4.2. Screening of MARs

Eleven aliquots of 30 mL of extract solutions (5.0 mg/mL) and each type of treated MARs (1 g dry resin) were added into conical flasks (100 mL), which were shaken continually for 24 h at 120 rpm and 25 °C. After filtration, the supernatants were used to analyze the concentrations of TFs, and the solid resins were cleaned with distilled water, then immediately desorbed with 30 mL of 95% ethanol under the same conditions for 24 h. The adsorption capacity, desorption capacity and desorption ratio were calculated based on the following equations:(1)$$q_{e}=\left(C_{0}-C_{e}\right)V_{i}/W$$
(2)$$q_{d}=C_{d}V_{d}/W$$
(3)$$D=q_{d}/q_{e}\times 100\%$$
where q_e_ and q_d_ stand for the adsorption capacity and desorption capacity; C_0_, C_e_ and C_d_ are the initial, the equilibrium and desorbed concentration of TFs (mg/mL), respectively; V_i_ and V_d_ represent the original sample volume and desorption solution volume (mL), respectively; W is the dry weight of resins used (g).

#### 3.4.3. Dynamic Analysis of MARs

The selected resins (HPD-300, HP20) were packed into the glass columns (200 × 10 mm). Sample solution (30 mL) was loaded into the column at a rate of 2 BV/h. After adsorption equilibrium, the flavonoids desorption was performed successively with distilled water (240 mL) and 95% ethanol (300 mL) at the flow rate of 4 BV/h. Then, the effluent was collected and dried to determine the recovery rate and purity of purified extracts, which was calculated according to the following equations:(4)$$Q_{R}=C_{2}\times V_{2}/C_{1}\times V_{1}\times 100\%$$
(5)$$Q_{p}=C\times V/M_{0}\times 100\%$$
where Q_R_ and Q_p_ (%) refer to recovery and purity of the flavonoids; C_1_ (mg/mL) is the equilibrium concentration of flavonoids in the sample solution; C_2_ and C (mg/mL) represent the concentrations of flavonoids in the desorption solution; V_1_ (mL) is the sample volume; V_2_ and V (mL) stand for the desorption volume; M_0_ (mg) is quality of the dry purified extract.

#### 3.4.4. Procedures for Chromatographic Purification

To obtain the optimum purification process, dynamic adsorption and desorption experiments were carried out on glass columns (200 × 10 mm) packed with HP20 resin in a packed bed volume (BV) of 18 mL. The extraction solution (2.54 mg/mL of TFs) was applied to the column bed with different feed rates (1, 2, 3 and 4 BV/h), and the adsorption capacity for TFs in the resin column was used to select the best flow rate. After the adsorption equilibrium, the flavonoids-loaded resin column was washed by distilled water to remove water-soluble impurities and immediately eluted by 10 BV of 10–60% ethanol at a constant flow rate of 2 BV/h. The best ethanol concentration was chosen based on purity of flavonoids. Subsequently, the sample-loaded columns were eluted by the optimal eluting agent at various flow rates (1, 2, 3 and 4 BV/h), and the eluted fractions were collected and analyzed.

### 3.5. UHPLC-ESI-QTOF-MS/MS Analysis

An Agilent 1290 Infinity UHPLC system (Santa Clara, CA, USA), connected with Agilent Plus C18 RRHD column (2.1 mm × 100 mm, 1.8 μm), was used to analyze flavonoids in the purified-extract sample (2.0 mg/mL). The mobile phase consisted of 0.1% formic acid in water (A) and acetonitrile (B). The elution steps used were as follows: 0–5 min, 5–10% B, 5–25 min, 10–15% B; 25–30 min, 15–30% B; 30–40 min, 30–95% B. The temperature of the column was maintained at 30 °C and injection volume was 2.0 μL.

Mass spectrometry was performed using an Agilent 6530 Q-TOF Mass Spectrometer (Santa Clara, CA, USA), equipped with an ESI interface operating in negative ion mode. The optimized MS parameters were as follows: gas temperature 350 °C, gas flow 10 L/min, capillary voltage 3000 V, skimmer voltage 65 V. The analyzed full-scan mass range was 50–1500 Da. The targeted-MS/MS collision energy (CE) was set at 40 eV. A reference solution was sprayed as continuous calibration with the following reference masses: *m/z* 112.9856 and 1033.9881.

### 3.6. In Vitro Antioxidant Capacity

The antioxidant assays of crude and purified extracts were evaluated at a series of concentrations (6.25–500 μg/mL), then determined by UV-vis spectrophotometer. The positive and negative control were, respectively, different concentrations of Trolox and methanol.

#### 3.6.1. DPPH• Scavenging Assay

The DPPH• scavenging activity was measured as described previously by [46] with a minor change. Briefly, 0.5 mL of DPPH methanolic solutions (0.1 mM) were added to tubes containing sample solutions (0.5 mL) with a series of concentrations. After reaction for 30 min in the dark, the absorbance of DPPH• was recorded at 517 nm. The DPPH• scavenging effect of each sample was calculated as follows:(6)$$\mathrm{DPPH}\;\mathrm{radical}\;\mathrm{scavenging}\;\mathrm{rate}=\left(1-A_{\mathrm{sample}}/A_{\mathrm{control}}\right)\times 100\%$$
where $A_{\mathrm{control}}$ and $A_{\mathrm{sample}}$ represent the absorbance of negative control and tested samples, respectively.

#### 3.6.2. ABTS•^+^ Scavenging Assay

The ABTS•^+^ scavenging activity of samples was monitored as previously reported by [47] with a minor modification. A total of 7.0 mM ABTS•^+^ solution and 2.45 mM K_2_S_2_O_8_ were mixed to prepare a stock solution of ABTS•^+^ which was stored in the dark for 16 h before use. Fresh ABTS•^+^ working solution (1.8 mL), that the stock solution of ABTS•^+^ was diluted in distilled water until its absorbance of 0.70 ± 0.01 at 734 nm, was added to 0.2 mL samples, and the mixture was shaken vigorously. After 10 min in the dark, the absorbance of ABTS•^+^ was measured at 734 nm. The ABTS•^+^ scavenging activity was estimated by the following equation:(7)$$\mathrm{ABTS}\;\mathrm{radical}\;\mathrm{cation}\;\mathrm{scavenging}\;\mathrm{rate}=\left(1-A_{\mathrm{sample}}/A_{\mathrm{control}}\right)\times 100\%$$
where $A_{\mathrm{control}}$ and $A_{\mathrm{sample}}$ are the absorbance of negative control and tested samples.

### 3.7. Cell Culture and Antiproliferative Activity Assay

The HepG2, MCF-7, Caco-2 and A549 cell lines were cultured in DMEM medium supplemented with 10% FBS, 100 U/mL penicillin and 100 μg/mL streptomycin at 37 °C (5% CO_2_, 95% relative humidity). These cells were seeded into 96-well microplates at a density of 1 × 10^5^ cells/mL and incubated overnight. After being treated with 200 μL of various concentrations of samples (25–500 μg/mL) for 48 h, 30 μL MTT (5.0 mg/mL) was added into each well for 4 h. Then, discarding the supernatants, to each well we added 200 μL of DMSO to dissolve the formazan crystals. The absorbance (OD) values were measured at 492 nm by a microplate reader (MultiskanTM FC, Thermo Fisher, Waltham, MA, USA).
(8)$$\mathrm{Cell}\;\mathrm{proliferation}\%={\mathrm{OD}}_{\mathrm{sample}}/{\mathrm{OD}}_{\mathrm{control}}\times 100\%$$

### 3.8. Assay for Antimicrobial Activity

The antimicrobial potentials of crude extracts and purified extracts were evaluated on Gram-positive bacteria (*S. aureus* (ATCC 6538), *S. epidermidis* (ATCC 12228)), Gram-negative bacteria (*E. coli* (ATCC 8739), *P. aeruginosa* (ATCC 27853)) and yeast (*C. albicans* (ATCC 90028)). Sample solutions were prepared with crude and purified extracts diffused in 50% methanol.

#### 3.8.1. Inhibition Zone Diameter Determination

Agar well diffusion method was used to measure the inhibition zone (ZOI) diameter. Bacterial/yeast suspension was added to the sterilized TSA (final bacterial/yeast concentration was about 10^6^ CFU/mL). Wells were made on the agar plates with sterile borer (6 mm) and then we added 200 μL sample solution (50 mg/mL). The positive and negative controls were, respectively, 50 μg/mL kanamycin and 50% methanol. After culturing for 24 h at 37 °C, the diameter of the inhibition zone was determined by the cross method.

#### 3.8.2. Measurement of Minimum Inhibition Concentration

Agar dilution method with minor modifications was used to estimate the minimum inhibition concentration (MIC) [48]. Bacterial/yeast suspensions (10^8^ CFU/mL) mixed with agar medium (1:999, *v/v*) were poured into petri dishes. Sample solution (200 μL) with different concentrations (1.56–50 mg/mL) were added in the wells (6 mm), with 50% methanol as negative control. The plates were cultured overnight at 37 °C.

### 3.9. Statistical Analysis

Values of all experiments were measured in triplicate and were expressed as the mean ± standard deviation (SD). Using one-way analysis of variance (ANOVA), statistical significances were analyzed with the software SPSS 23.0 (SPSS Inc., Chicago, IL, USA). Multiple comparison of means was calculated according to Duncan’s multiple range test. P-values less than 0.05 were considered statistically significant.

## 4. Conclusions

The present study investigated the purification of flavonoids in *Rosa setate x Rosa rugosa* waste, which provides an effective and feasible method for the exploitation of waste biomass resource. The results of static and dynamic screening analysis indicated that HP20 macroporous resin was the most suitable for purifying total flavonoids from *R. setate x R. rugosa* wastes. The optimum parameters for the enrichment process with HP20 resin were as follows: HP20 resin column was loaded with 4 BV sample (containing 2.54 mg/mL of TFs) at 2 BV/h flow rate and desorbed by 9 BV of 30% ethanol at a rate of 3 BV/h. After one-step enrichment, there was a 10.7-fold increase in the purity of flavonoids with a recovery yield of 82.02%. A total of 14 flavonoids were tentatively identified through UHPLC-QTOF-MS technology, including isorhamnetin-3-*O*-sophoroside, quercetin derivatives (quercitrin, rutin, hyperoside, etc.) and kaempferol derivatives (astragalin, kaempferol-3,4′-di-*O*-diglucoside, etc.). In addition, this is the first evaluation of the biological activities of purified extracts from Kushui rose waste. Bioassay results demonstrated that flavonoids-enriched extracts had superior antioxidant, antiproliferative and antibacterial activities compared with crude extracts. These findings suggested that *R. setate x R. rugosa* waste is a novel source of bioactive flavonoids and the purified product could be used to develop promising active ingredients in dietary supplements or pharmaceuticals. Further studies should focus on the isolation of main flavonoid molecules and their biological action mechanisms.

## Figures and Tables

**Figure 1 molecules-27-04379-f001:**
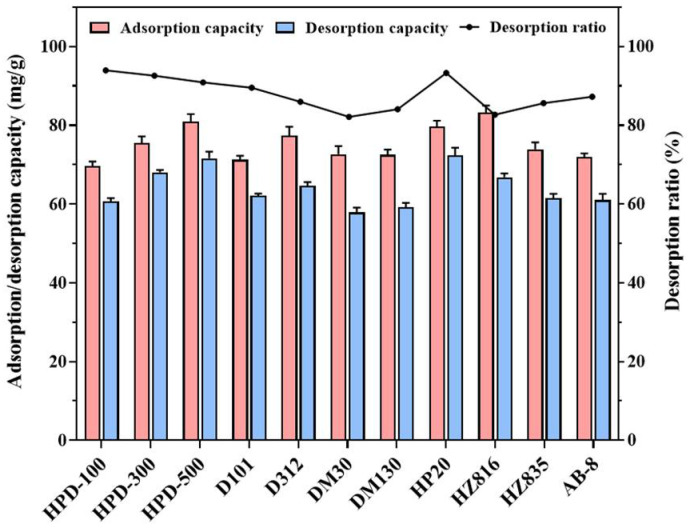
Static screening results of total flavonoids in *Rosa setate x Rosa rugosa* waste extracts on eleven macroporous adsorption resins.

**Figure 2 molecules-27-04379-f002:**
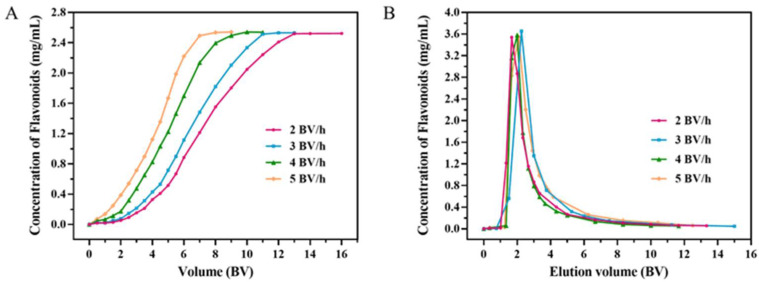
Dynamic breakthrough curves (**A**) and dynamic desorption curves (**B**) of TFs in crude extracts on HP20 resin column at various flow rates.

**Figure 3 molecules-27-04379-f003:**
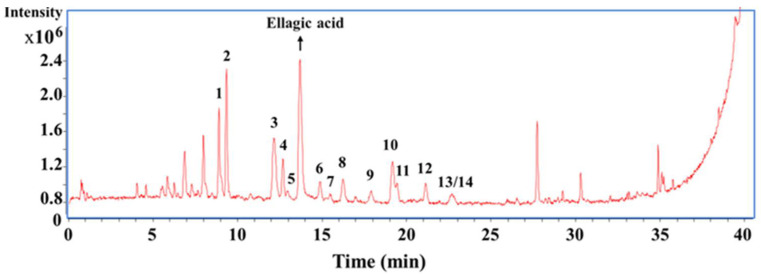
Base-peak chromatogram (in negative ion mode) of the purified extracts from *Rosa setate x Rosa rugosa* wastes.

**Figure 4 molecules-27-04379-f004:**
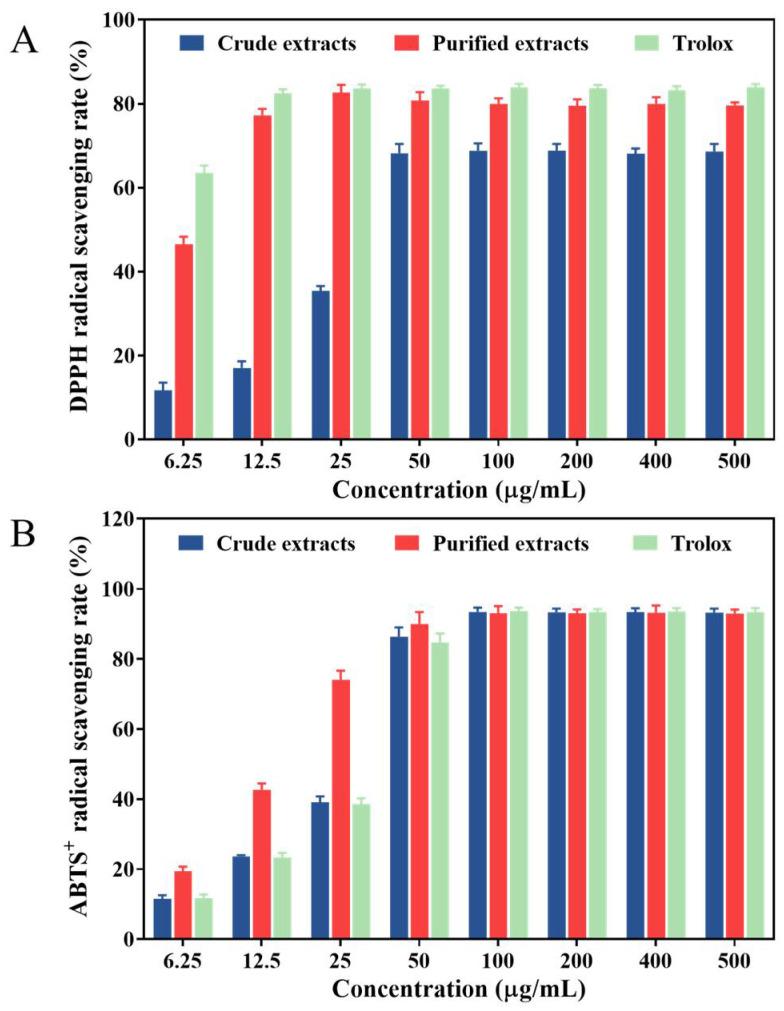
Antioxidant activities of crude and purified extracts in vitro. DPPH• scavenging activity (**A**) and ABTS•^+^ scavenging activity (**B**).

**Figure 5 molecules-27-04379-f005:**
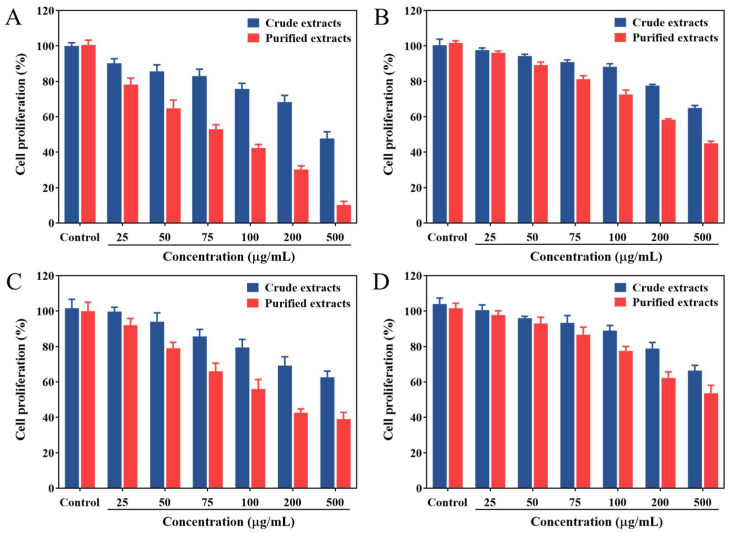
Antiproliferative activities of crude extracts and purified extracts from *Rosa setate x Rosa rugosa* waste against HepG-2 (**A**), MCF-7 (**B**), Caco-2 (**C**) and A549 (**D**) cell lines.

**Figure 6 molecules-27-04379-f006:**
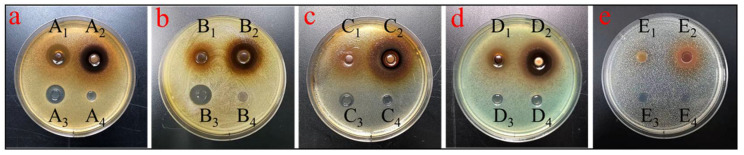
Effect diagram of inhibition zone diameter for *S. aureus* (**a**), *S. epidermidis* (**b**), *E. coli* (**c**), *P. aeruginosa* (**d**) and *C. albicans* (**e**). Crude extracts (A1, B1, C1, D1, E1); purified extracts (A2, B2, C2, D2, E2); kanamycin (A3, B3, C3, D3, E3); 50% methanol blank control (A4, B4, C4, D4, E4).

**Table 1 molecules-27-04379-t001:** Flavonoids identified in purified extracts from *Rosa setate x Rosa rugosa* waste using UHPLC-QTOF-MS/MS.

Peak	t_R_ ^a^ (min)	[M-H]^−^ (*m/z*)	Formula	MS/MS Fragment Ion (*m/z*)	Tentative Identification
1	8.900	625.1449	C_27_H_30_O_17_	300	Quercetin-3,4′-di-*O*-diglucoside
2	9.350	625.1462	C_27_H_30_O_17_	445, 300	Quercetin-3-*O*-sophoroside
3	12.168	609.1463	C_27_H_30_O_16_	285, 284	Kaempferol-3,4′-di-*O*-diglucoside
4	12.701	639.1613	C_28_H_32_O_17_	477, 315, 314, 300, 299	Isorhamnetin-3-*O*-sophoroside
5	12.951	609.1479	C_27_H_30_O_16_	489, 429, 327, 284	Kaempferol-3-*O*-sophoroside
6	14.902	463.0921	C_21_H_20_O_12_	301, 300, 271, 255	Hyperoside ^b^
7	15.486	609.1476	C_27_H_30_O_16_	301, 300, 271	Rutin ^b^
8	16.253	463.0924	C_21_H_20_O_12_	301, 300, 271, 255	Isoquercitrin ^b^
9	17.920	433.0799	C_20_H_18_O_11_	301, 300, 271, 255	Quercetin-3-*O*-pentoside
10	19.187	615.1020	C_28_H_24_O_16_	463, 301, 179, 151	Quercetin-3-*O*-(6″-galloyl)-glucoside
11	19.437	579.1440	C_26_H_28_O_15_	284, 255, 227	Kaempferol-3,7-di-*O*-rhamnoside
12	21.138	433.0796	C_20_H_18_O_11_	301, 300, 271, 255	Quercetin-3-*O*-pentoside
13	22.589	447.0962	C_21_H_20_O_11_	285, 284, 255, 227	Astragalin
14	22.789	447.0974	C_21_H_20_O_11_	301, 300, 271, 151	Quercitrin

^a^ Retention time on UPLC analysis. ^b^ Compound confirmed by authentic standard.

**Table 2 molecules-27-04379-t002:** Zone of inhibition (ZOI) and minimum inhibition concentration (MIC) of crude extracts and purified extracts from *Rosa setate x Rosa rugosa* waste against tested strains.

Experimental Strains	Crude Extracts		Purified Extracts		Kanamycin
	ZOI ^a^ (mm) 50 mg/mL	MIC ^b^ (mg/mL)	ZOI ^a^ (mm) 50 mg/mL	MIC ^b^ (mg/mL)	ZOI ^a^ (mm) 50 μg/mL
*Staphylococcus aureus*	16.8 ± 0.24	12.5	20.7 ± 0.4	1.5625	17.2 ± 0.29
*Staphylococcus epidermidis*	11.9 ± 0.15	12.5	18.9 ± 0.1	1.5625	16.9 ± 0.33
*Escherichia coli*	9.5 ± 0.38	25.0	13.8 ± 0.23	6.25	11.9 ± 0.26
*Pseudomonas aeruginosa*	12.2 ± 0.13	12.5	16.4 ± 0.22	3.125	-
*Candida albicans*	-	-	11.2 ± 0.17	12.5	-

^a^ Zone of inhibition. ^b^ Minimum inhibition concentration. -, Undetectable.

## Data Availability

The authors declare that all data supporting the findings of this study are available within the article.

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
