# Peer review of "Purification and Identification of Flavonoid Molecules from Rosa setate x Rosa rugosa Waste Extracts and Evaluation of Antioxidant, Antiproliferative and Antimicrobial Activities"

_molecules, 2022, doi:10.3390/molecules27144379_

Round 1
Reviewer 1 Report
Manuscript "Purification of flavonoids from Rosa setate x Rosa rugosa waste extracts and evaluation of antioxidant, antiproliferative and antimicrobial activities" by Wu et al. explains that flavonoids-enriched extracts from R. setate x R. rugosa waste have the potential to be applied for functional food and pharmaceutical industries. It is well-written and well-presented manuscript of interest for the readers of Molecules. Few minor comments: please increase the font size in Figure 1, and expand figure legends in Fig. 4 and 5.
Author Response
Dear reviewer:
On behalf of my-coauthors, we are greatly appreciated that you have give some suggestions for improving our work. The comments are valuable and helpful for us. We have considered comments carefully and have tried our best to make revision.

Reviewer 2 Report
Wu et al. investigated an interesting research titled “Purification of flavonoids from Rosa setate x Rosa rugosa waste extracts and evaluation of antioxidant, antiproliferative and antimicrobial activities”. Although the research is scientifically sound, it must be revised to solve some important issues before it can be considered for publication in Molecules journal.
Title
I would advise the author to alter the manuscript title to something more appropriate to the content, as mentioned below.
“Purification and identification of flavonoid molecules from Rosa setate x Rosa rugosa waste extracts and evaluation of antioxidant, antiproliferative and antimicrobial activities”
Abstract
1. Include a statement describing why antioxidant, antiproliferative, and antibacterial activities were chosen to be performed.
2. Remove the word "activity" from the keywords and update it as "antioxidant," "antiproliferative," and "antimicrobial."
Introduction
1. What do you mean by "little research" on lines 40-42? Modify the sentence.
2. The importance of flavonoids purification was only mentioned in the introduction. However, the authors said nothing about the biological properties reported on Rosa setate x Rosa rugose's, including antioxidant, antiproliferative, and antibacterial activity.
3. What do you mean by "few researchers" on line 56? Make a change to the sentence. Instead, say "Previous studies have demonstrated..." Also, make sure to include any relevant references.
4. Consider including prior study limitations, along with any relevant references, and how current flavonoids purification research addresses them.
5. Throughout the manuscript, avoid using the phrases "I," "we," or "our."
Results
1. This section's title should be modified to Results and Discussion. Because, the author includes a discussion part in this section.
2. 14/Fourteen flavonoids in Section 2.3? Maintain a consistent format throughout the manuscript.
3. Table 1, include the chemical structure of flavonoids that have been identified.
4. Which of these 14 flavonoids is most likely to be responsible for biological action? Authors must do a literature review and report their findings with appropriate references.
5. In section 2.6, the full name and abbreviations of microorganisms should be explicitly specified before using abbreviated words.
6. Line 239 is meant to represent Table 2. Please check.
7. The legend in Figure 6 needs to be improved.
8. In Table 2 title, Antibacterial (MIC)?
9. I believe the authors should refocus their discussion to show how the results of their investigation fit into the larger picture of current issues including antibiotic resistance.
10. Key justifications for the findings must be presented by the author.
Materials and Methods
Overall, it is well-written, however all of the sub-sections' procedures are excessively long. Wherever possible, simplify some of the information and make it very clear.
Conclusions
The author should continue to improve the sentence about work's novelty. The sentence on future perspectives based on antioxidant, antiproliferative, and antibacterial activities needs to be added in the conclusion.
Author Response
Dear reviewer:
On the behalf of my co-authors, we are greatly appreciated that you have given some positive comments and suggustions for improving our work. We have considered comments carefully and tried our best to make revisions. Detailed corrections are listed below point by point marked in red.

Round 2
Reviewer 2 Report
In response to reviewer comments, the authors implemented amendments to the manuscript. Therefore, I recommend for publication in Molecules in its present form.